# COVID-19 Vaccine—A Potential Trigger for MOGAD Transverse Myelitis in a Teenager—A Case Report and a Review of the Literature

**DOI:** 10.3390/children9050674

**Published:** 2022-05-06

**Authors:** Cristina Oana Mărginean, Lorena Elena Meliț, Maria Teodora Cucuiet, Monica Cucuiet, Mihaela Rațiu, Maria Oana Săsăran

**Affiliations:** 1Department of Pediatrics I, George Emil Palade University of Medicine, Pharmacy, Science, and Technology of Târgu Mures, Gheorghe Marinescu Street No. 38, 540136 Târgu Mureș, Romania; marginean.oana@gmail.com; 2Faculty of Medicine, George Emil Palade University of Medicine, Pharmacy, Science, and Technology of Târgu Mures, Gheorghe Marinescu Street No. 38, 540136 Târgu Mureș, Romania; teacucuiet@gmail.com; 3Pediatric Neuropsychiatry County Emergency Hospital Târgu Mureș, Gheorghe Marinescu Street No. 50, 540136 Târgu Mureș, Romania; monicacucuiet@yahoo.com; 4Department of Radiology, George Emil Palade University of Medicine, Pharmacy, Science, and Technology of Târgu Mures, Gheorghe Marinescu Street No. 38, 540136 Târgu Mureș, Romania; d_a_mihaela@yahoo.com; 5Department of Pediatrics III, George Emil Palade University of Medicine, Pharmacy, Science, and Technology of Târgu Mureș, Gheorghe Marinescu Street No. 38, 540136 Târgu Mureș, Romania; oanam93@yahoo.com

**Keywords:** COVID-19 vaccine, MOGAD transverse myelitis, teenager

## Abstract

MOGAD-transverse myelitis is a rare disorder in children and adults, but with a higher incidence in pediatric patients. We report a case of MOGAD-transverse myelitis in a boy who was admitted to hospital with bilateral motor deficit of the lower limbs associated with the impossibility of defecating and urinating. The symptoms progressively developed with severe fatigue within the week prior to admission, with the impossibility to stand occurring 36 h before admission. The anamnesis found that he was vaccinated for COVID-19 approximately 6 weeks before admission to our clinic. The laboratory tests revealed a normal complete cellular blood count, without any signs of inflammation or infection, except for both cryoglobulins and IgG anti-MOG antibodies. MRI showed a T2 hypersignal on vertebral segments C2-C5, Th2-Th5 and Th7-Th11, confirming the diagnosis of longitudinally extensive transverse myelitis. The patient received intravenous high-dose methylprednisolone (1 g) for 5 days, associated with prophylactic antibiotic treatment, subcutaneous low-molecular-weight heparin and other supportive treatment. The patient was discharged on the 12th day of admission, able to walk without support and with no bladder or bowel dysfunction. We can conclude that an early diagnosis was essential for improving the patient’s long-term outcome.

## 1. Introduction

The myelin oligodendrocyte glycoprotein (MOG) is produced by the myelin-forming cells of the central nervous system (CNS), and it was first discovered in the late 1970s [1]. The human mature MOG contains a signal peptide comprising 29 amino acids which is completed by 218 amino acids of the mature protein [2]. This protein is a member of the immunoglobulin superfamily and consists of an extracellular immunoglobulin variable (IgV) domain; two hydrophobic regions: a transmembrane domain and a domain within the membrane bilayer; a short cytoplasmic loop; and a cytoplasmic end [1]. Despite the fact that the MOG is encountered only in relatively low amounts within myelin, it can be easily attacked by possible antibodies and T-cell response due to its extracellular IgV domain and its location at the end of the myelin sheaths [1]. The MOG seems to be involved in regulating the stability of the oligodendrocyte microtubule, mediating the interactions between myelin and the immune system, and maintaining the integrity of the myelin sheaths structure [1]. Over the past few years, multiple studies focused on assessing the role of autoantibodies against MOGs (MOG-abs) acting as potential biomarkers for demyelinating CNS diseases [3,4]. Initially, these antibodies were thought to be included in the wide spectrum of biomarkers used for the diagnosis of multiple sclerosis, but they were only identified in a minor group of patients with multiple sclerosis [5,6,7]. Therefore, it became widely accepted that MOG-abs in fact indicate a different disease and could confirm the diagnosis of multiple sclerosis [6,8]. MOG-abs have invariably been detected in a wide spectrum of demyelinating syndromes, predominantly in pediatric patients [9,10]. 

The demyelinating spectrum of disorders triggered by MOG-abs includes transverse myelitis (TM) and longitudinally extensive transverse myelitis (LETM), but also acute disseminated encephalomyelitis (ADEM), brainstem or cortical encephalitis, and unilateral/bilateral optic neuritis (ON), recently referred to as MOG-antibody disease (MOGAD). In spite of its rarity, the prevalence of MOGAD was reported to be higher in pediatric patients, accounting for 40% of the cases when compared to mixed cohorts, while adult samples accounted for 29% of cases with a 22% prevalence [11]. These findings were also sustained by a recent Dutch study that found an incidence of 0.31/100.000 per year for children in comparison to 0.13/100.000 per year for adults [10]. The most common clinical presentations of pediatric MOGAD at onset include ADEM—46% of the cases, ON—30% of the cases, TM—11% of the cases, and simultaneous ON and TM—4% of the cases [12]. Myelitis is the second-most frequent presentation of MOGAD in adult patients accounting for 20% of disease-related attacks, while its prevalence is less common in pediatric MOGAD patients [10,13,14]. LETM is defined as a spinal cord injury spanning at least three vertebral segments in length being a specific finding in MOGAD [15]. The symptoms consist of motor and/or sensory deficits, as well as erectile, bladder and/or bowel dysfunction [15]. Multiple clinical differences were proposed for distinguishing between myelitis in MOGAD patients and those with multiple sclerosis or aqua porin 4 neuromyelitis optica spectrum disorders, such as a tendency to especially affect younger males, increased frequency of erectile and bladder dysfunction, prodromal infection, and simultaneous ADEM [16]. In terms of diagnosis, magnetic resonance imaging remains the most useful tool since it reveals abnormalities in the spinal cord, brain and/or the optic nerve according to the clinically impaired anatomical area of the nervous system [17]. The MRI in patients with LTEM, which especially affects grey matter, revealed an ‘H-sign’ on the axial plane [15]. Only 25% of MOGAD myelitis cases presented a gadolinium contrast-enhancement of the spinal cord injuries [15]. Moreover, it was reported that 10% of the patients diagnosed with MOGAD displaying myelitis attacks might initially present with normal spinal cord MRI [18]. Nevertheless, these patients present no specific clinical and imaging findings. Thus, the positive MOG-IgG serum levels remain the most reliable diagnostic tool for the confirmation of MOGAD in patients with clinical or neuroimaging findings suggesting a demyelinating disorder. The serum levels of MOG-IgG are related to disease activity, with an increased concentration throughout acute attacks and decreased or even absent concentration in the setting of remission, during the chronic phase, or following a monophasic incident [19]. Taking into account that treatment might also influence the levels of these antibodies, the patients should be retested within the following 1–3 months after plasma exchange, steroids or immunoglobulin infusion [20]. 

Based on the paucity of MOGAD cases, there are no evidence-based treatment guidelines for patients with MOGAD, but similar treatment options as in multiple sclerosis and neuromyelitis optica spectrum have been attempted so far in this group of patients. The election therapy for the acute attacks involves high-dose intravenous methylprednisolone (0.5–2 g for 5–10 days), but other therapeutic options include plasma exchange and intravenous immunoglobulin in patients with more severe attacks or those who are unresponsive to steroids [1]. 

We describe a case of MOGAD-transverse myelitis in a male teenager in order to highlight the importance of urgent diagnosis and treatment to improve the patient’s outcome. Informed consent from the patient’s mother was obtained prior to the publication of this case.

## 2. Case Report

### 2.1. Presenting Concerns

We present the case of a 15-year-old male teenager admitted to our clinic for bilateral motor deficit of the lower limbs associated with the impossibility to defecate for 48 h prior to admission and to urinate for approximately 10 hours. The symptoms developed progressively with severe fatigue within the last week and the impossibility to stand occurring approximately 36 hours before admission. The anamnesis found that he was vaccinated for COVID-19 approximately 6 weeks before admission to our clinic, but without any signs of infection prior to the onset of these symptoms. 

### 2.2. Clinical Findings

The clinical exam on admission found motor and sensorial deficits of the lower limbs associated with bladder and bowel disfunction. The neurological exam was performed in dorsal decubitus while the patient was conscious, oriented in time and space regarding his own person, but he could not sit or stand. The patient did not present any signs of meningeal or cranial nerves impairment. The examination of the upper limbs revealed no impairment of the tonus or muscular strength, with equal and symmetrical reflexes, while the examination of the lower limbs found severe and symmetric motor deficits, with bilateral hypotonia, exacerbated reflexes, and the presence of Babinski signs. The cutaneous abdominal reflexes were absent, and the patient presented with a retention-type impairment of vesical and anal sphincters. The neurological exam of the spine was normal. The patient weighed 62 kg. 

### 2.3. Diagnostic Focus and Assessment

The routine laboratory tests performed on the first day of admission found a normal and complete cellular blood count without any signs of inflammation or infection. The liver and kidney function parameters were within normal limits. Moreover, the creatin kinase and lactate dehydrogenase levels were normal. The serology for Ebstein Barr, Toxoplasma, Rubella, Cytomegalovirus, Herpes virus 1 and 2, as well as Borellia spp was negative. The antibodies for Hepatitis C and the antigen HBs were also negative. Other tests revealed no deficits of vitamin B9, B12 or D. The cranial computed tomography on the day of admission found no pathological findings. We performed a cranial and lumbar MRI, which showed a T2 hypersignal on vertebral segments C2-C5, Th2-Th5 and Th7-Th11, confirming the diagnosis of longitudinally extensive transverse myelitis (Figure 1).

We performed a lumbar puncture, but the cytology exam of the spinal fluid was normal with no biochemical signs of infection and a negative culture. Other tests from the spinal fluid, such as oligoclonal bands and albumin, were negative, except for the IgG level, which was mildly positive: 48 mg/L (normal values < 34 mg/L). All serological tests that were performed to rule out the causes of autoimmune disorders—anti-aquaporin 4 antibodies, antinuclear antibodies, anti-DNA antibodies, anti-cardiolipin, citrullinate citric peptide, immune circulating complexes, C3 and C4 fractions of the complement, lupus anticoagulant, and IgM anti-phospholipids antibodies—were either negative or within normal limits, except for cryoglobulins, which were positive. Moreover, the IgG anti-MOG antibodies were positive, 1:60 (normal range < 1:10). Therefore, we established the diagnosis of MOGAD-longitudinally extensive transverse myelitis of the cervical and thoracal spinal cord.

### 2.4. Therapeutic Focus and Assessment

We initiated high-dose methylprednisolone (1 g) intravenously for 5 days, which was associated with prophylactic antibiotic treatment according to an infectious diseases specialist, subcutaneous low-molecular-weight heparin, considering the immobilization and other supportive treatments, such as crystalloid solutions and vitamins. We also inserted a bladder catheter immediately after admission and performed an enema each day to relieve bladder and bowel dysfunctions. The motor and sensorial deficits improved after approximately 3 days of treatment, and the patient was able to stand on the fifth day of admission. Thus, we gradually tapered the steroids dose and removed the urinary catheter. The patient was discharged on the 12th day of admission being able to walk without support and with no bladder or bowel dysfunction.

### 2.5. Follow-up and Monitoring

We repeated the MOG-abs after approximately 3 months, and they remained positive, albeit with a mild decrease (1:40). We also performed a control MRI that revealed the minimum hypersignal of the C3-C5 segments (Figure 2). The clinical status of the patient was normal. 

## 3. Discussions

MOGADs have a low incidence of approximately 1.1–2.4/million people [21]. The incidence of these disorders in pediatric patients is 3.1 per million children, considerably higher in comparison to adult populations [10]. Isolated TM might occur in approximately 20% of the patients with MOGAD but might also be associated with ON in an additional 8–15% of the cases [13,22,23]. Moreover, an association between TM and ADEM was also reported in 17% of the patients [15]. Our case presented no clinical or neuroimaging signs of ON or ADEM. Similar to aquaporin 4-positive neuromyelitis optica spectrum disorders (AQP4-positive NMOSD), the onset of MOGAD seems to be related to a previous acute infection or vaccination [24]. Thus, during the global pandemic, a series of MOGAD cases related to SARS-CoV-2 infection were reported [25,26,27]. In addition, a recent reported case described MOGAD encephalomyelitis in a 43-year-old woman following COVID-19 vaccination [28]. Similarly, occasional cases of inflammatory CNS disorders were also described following COVID-19 vaccination, such as seronegative NMOSD-like associated with multiple sclerosis, but also a patient with AQP4-positive NMOSD [29,30]. MOGAD-LETM occurred in our patient approximately 6 weeks after his second dose of COVID-19 vaccine (Pfizer), suggesting that this vaccine might have an effect on the occurrence of the acute attack in this case since we did not find any other triggers. 

A recent retrospective study included 54 patients with MOGAD-transverse myelitis, among which 16 (30%) were children with isolated TM (54%) or TM combined with ADEM or ON [15]. Patients with isolated TM or LETM usually presented with bilateral motor and sensory deficits involving bladder and bowel dysfunction and developing within a range from hours to days [31]. Moreover, one-third of these patients might be wheelchair bound [15]. In addition, erectile dysfunction was associated with 54% of the patients included in the aforementioned study [15]. In our case, the motor and sensory deficits involved both lower limbs, as well as bladder and bowel dysfunction, progressing within the last 3–5 days, but our patient did not develop any signs of erectile dysfunction. Erectile dysfunction might be attributed to the longitudinally extensive lesions involving conus, which was observed by MRI in up to 41% of the cases [15]. The involvement of the conus was also reported in 37% of all pediatric patients with MOGAD who presented with spinal cord impairment [32], being more typical for MOGAD-TM, albeit reported in patients with AQP4-positive TM [15,33,34]. Moreover, the flaccid areflexia, which was also present in our case, as well as in 19% of the cases included in the previously mentioned retrospective study, might be explained by the predilection of MOG-abs for the central grey matter, similar to AQP4-positive TM, but atypical for multiple-sclerosis TM [15]. 

The diagnosis of TM or LTEM should be suspected in cases presenting with the above-mentioned clinical picture and positive MRI signs, such as hyperintense ‘H-sign’ in the axial orientation associated with a narrow vertical line in the T2-weighed sagittal plane image. This suggests a major impairment of grey matter within the spinal cord in comparison to AQP4-positive TM, which might not appear as centrally located in the spinal cord [15,35]. Nevertheless, the final diagnosis should only be established in the presence of MOG-abs. Thus, not only the patients with an increased risk of MOGAD should benefit from MOGD-abs testing, but also patients who present with findings atypical for multiple sclerosis or NMSOD, or those with an NMSOD phenotype and AQP4-negative IgG [1]. 

Considering that MOGAD is a relatively recent and rare disease, with age-dependent differences in terms of onset symptoms, no evidence-based guidelines are available regarding acute and maintenance immunotherapy since large treatment trials have not yet been performed. Nevertheless, the studies reported so far in patients with MOGAD used similar therapeutic approaches to multiple sclerosis and NMOSD, with high-dose intravenous methylprednisolone for the management of acute exacerbations showing a high effectiveness with complete recovery in more than 50% of the subjects with ON, brainstem and cortical encephalitis [1,36,37]. Our patients also presented a full recovery after intravenous methylprednisolone. Nevertheless, several reports underlined a risk of relapsing after steroid cessation dose reduction [13,24,38,39]. Moreover, according to a trial performed in UK, the number of second relapses was considerable with a new attack in almost half of cases over 2 years [13]. Another study revealed an even higher relapse rate, occurring in approximately 80% of patients during follow-up [24]. It was stated that disability progression is relapse-dependent in MOGAD patients, suggesting that maintenance therapy should be mainly considered in patients with a persistent positive serology since studies proved a greater risk of a relapsing disease course in the setting of persistent seropositivity over time [39]. Maintenance therapy includes oral corticosteroids, rituximab, methotrexate, azathioprine, mycophenolate mofetil and repeated doses of intravenous immunoglobulin [22]. Based on persistent seropositivity after 3 months from the initial attack, we also intend to initiate maintenance therapy in our case, in spite of his outstandingly favorable evolution with full recovery. 

As far as we know this is the first case of MOGAD-TM in a pediatric patient following COVID-19 vaccination, which might be considered the most likely trigger for our patient.

## 4. Conclusions

MOGAD-transverse myelitis is generally a rare disorder in both children and adults, but it has a higher incidence in pediatric patients. Its early diagnosis is essential for improving a patient’s long-term outcome. Patients diagnosed with MOGAD should be closely monitored to ensure prompt treatment in the case of future relapses. 

## Figures and Tables

**Figure 1 children-09-00674-f001:**
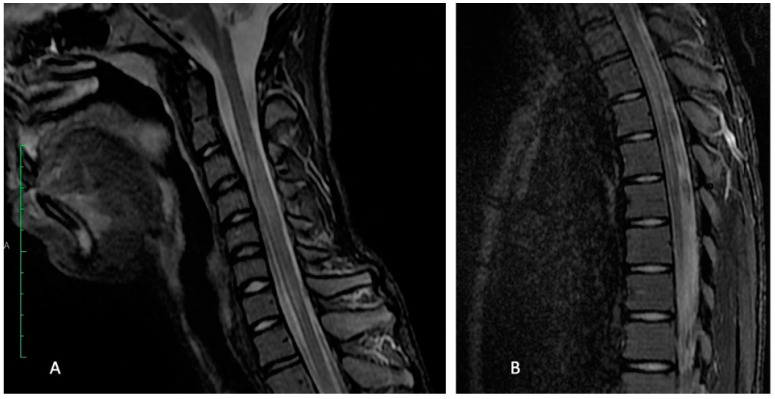
Sagittal STIR MRI at acute onset: intradural hyperintensity seen in the cervical (**A**) and dorsal (**B**) segments, with mild expansion of cord in cervical C3-C5 part.

**Figure 2 children-09-00674-f002:**
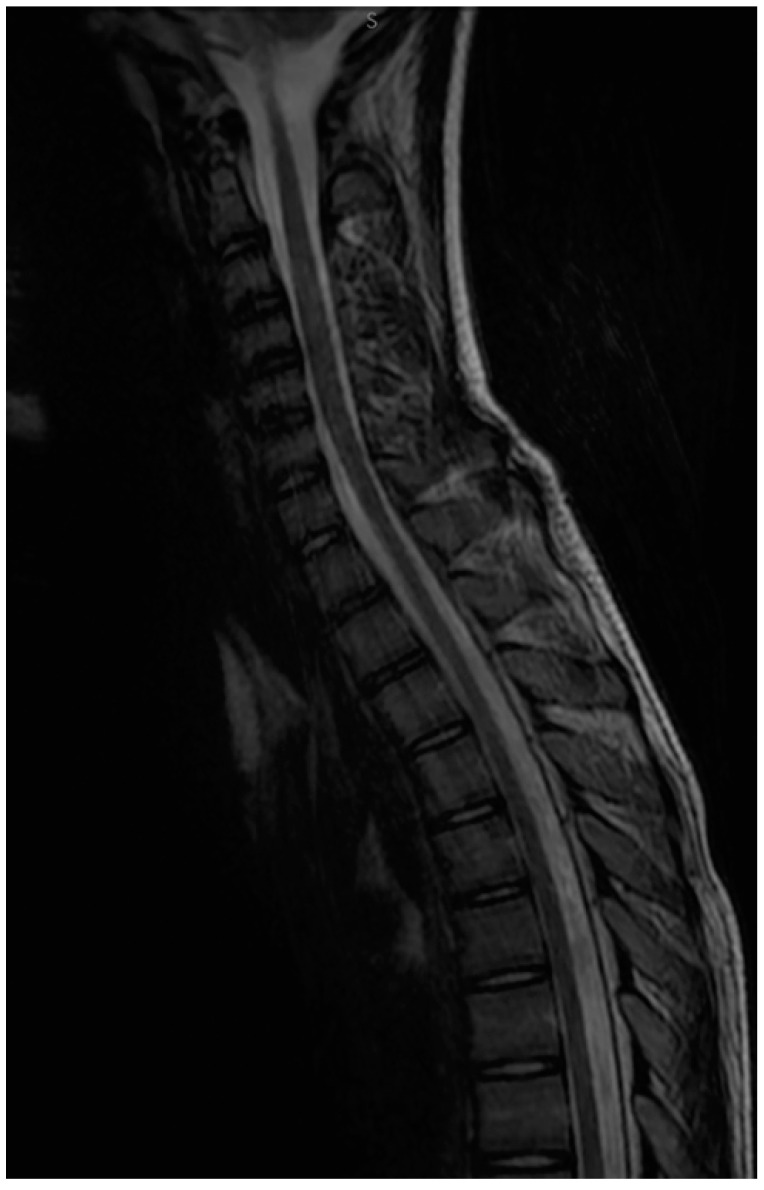
Sagittal T2 MRI at 3 months follow-up—slight remaining intradural hyperintensity seen in the cervical C3–C5 segment.

## Data Availability

Not applicable.

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
