# Peer review of "COVID-19 Vaccine—A Potential Trigger for MOGAD Transverse Myelitis in a Teenager—A Case Report and a Review of the Literature"

_children, 2022, doi:10.3390/children9050674_

Round 1

Reviewer 1 Report

Thank you fro the opportunity of reviewing this interesting paper regaridng a case of MOG-associated LTEM.

However, I have some minor suggestions.

Was there information about any infectious symptoms in the weeks prior to the onset of neurological symptoms?

Have you performed RT-PCR for bacteria and viruses, including SARS-CoV-2, on CSF? I think this is a fundamental diagnostic step to rule out a post-infectious mechanism. 

Could you explain in detail the neurological examination on admission?

Did you perform somatosensitive-evoked potentials?

Author Response

Reviewer 1

Comment 1

Thank you for the opportunity of reviewing this interesting paper regaridng a case of MOG-associated LTEM.

Answer 1

Thank you for your positive comments.

Comment 2

However, I have some minor suggestions.

Was there information about any infectious symptoms in the weeks prior to the onset of neurological symptoms?

Answer 2

We apologize for not mentioning this information in the initial form of our manuscript. However, we introduced the following information according to your recommendation in the revised form of our manuscript: ‘…but without any signs of infection prior to the onset of these symptoms’.

Comment 3

Have you performed RT-PCR for bacteria and viruses, including SARS-CoV-2, on CSF? I think this is a fundamental diagnostic step to rule out a post-infectious mechanism. 

Answer 3

Thank you for raising this issue. Nevertheless, we already mentioned that the patient had no previous recent history of any signs of infection. Moreover, we performed multiple cultures from the blood, urine, nose and throat, but we did not detect any signs of bacterial infection. We also performed a wide-range of serological test for multiple viruses which were all negative. Regarding the CSF, we found no signs on cytology and microbiology indicating any signs of infection. The culture from the CSF was also negative. We performed an RT-PCR from CSF for ruling out tuberculosis, which was negative. In terms of SARS-CoV-2 infection, we performed an RT-PCR exam from the nose and throat, which was negative and along with the fact that the patients was recently vaccinated with positive IgG antibodies, we did not consider that the cost of RT-PCR for SARS-CoV-2 from CSF were justified since this test in not available in our hospital and we had to be judicious regarding the diagnostic tools since the costs for most of the tests had to be supported by the patient’s parents. Therefore, we assessed thoroughly the necessity to perform each test and we strictly performed those that were indeed relevant for the diagnosis.

Comment 4

Could you explain in detail the neurological examination on admission?

Answer 4

According to your recommendations we introduced the following information in the section of Clinical findings: ‘The neurological exam was performed in dorsal decubitus while the patient was conscient, oriented in time and space and regarding his own person, but he could not sit or stand. The patient did not present any sings of meningeal or cranial nerves impairment. The exam of the upper limbs revealed no impairment of the tonus, muscular strength, with equal and symmetrical reflexes; while the exam of the lower limbs pointed out severe and symmetric motor deficit, with bilateral hypotonia, exacerbated reflexes, and the presence of Babinsky sign. The cutaneous abdominal reflexes were abolished and the patient present retention type impairment of vesical and anal sphincters. The neurological exam of the spine was normal.

Comment 5

Did you perform somatosensitive-evoked potentials?

Answer 5

We did not perform somatosensitive-evoked potentials since they were not available in our hospital and the patient was not transportable initially. Moreover, it is well-known that these potentials are usually normal in these patients, and they are not predictive for the outcome. These potentials are useful for the diagnosis of multiple sclerosis, and therefore, once we established the diagnosis they were not included in the workup of transverse myelitis.

Reviewer 2 Report

The paper is look well and worthy for publication. Every original case report must be available  for reader and Children is the excellent platform for its.   For my suggestion  the main question of this report is the side effect of COVID19. The authors indicated there important side effect that  COVID19 vaccine induced the myelitis in children.  This fact is well confirmed by authors therefore I suggested to publish its.

Author Response

Comment 1

The paper is look well and worthy for publication. Every original case report must be available  for reader and Children is the excellent platform for its.  

For my suggestion  the main question of this report is the side effect of COVID19. The authors indicated there important side effect that COVID19 vaccine induced the myelitis in children.  This fact is well confirmed by authors therefore I suggested to publish its.

Answer 1

We express our sincere gratitude for your review and the time you spent on assessing our manuscript. Thank you for your positive comments.

Reviewer 3 Report

The authors reported the case of MOGAD-transverse myelitis in a boy who was admitted at the hospital for bilateral motor deficit of the lower limbs. The manuscript is well written, with the review of the literature. However, I do not find sufficient evidence that the transverse myelitis was in direct corellation with the COVID vaccine. The only evidence is the information from the anamnesis pointing out that he was vaccinated for COVID-19 approximately 6 weeks before the admission to the clinic. I would suggest to the authors to confirm this relationship since this is highlighted in the article title.

Author Response

Reviewer 3

Comment 1

The authors reported the case of MOGAD-transverse myelitis in a boy who was admitted at the hospital for bilateral motor deficit of the lower limbs. The manuscript is well written, with the review of the literature.

Answer 1

Thank you for your positive comments.

Comment 2

However, I do not find sufficient evidence that the transverse myelitis was in direct corellation with the COVID vaccine. The only evidence is the information from the anamnesis pointing out that he was vaccinated for COVID-19 approximately 6 weeks before the admission to the clinic. I would suggest to the authors to confirm this relationship since this is highlighted in the article title.

Answer 2

Thank you for your valuable time spent on assessing our manuscript. As we already stated in the title we consider that the vaccine might have been a trigger, the most-likely trigger, therefore we mentioned ‘a potential trigger’ based on the fact that we did not identify any other causes related to the onset of MOGD-TM in our patient. Taking into account the well-documented hypothesis, which in fact was reported by previous authors, that MOGAD-TM might be preceded by an infectious stimulus or a vaccine, and the fact that we investigated our patient thoroughly without identifying any other potential causes for this condition, we considered to be highly important to report this case in order to increase the awareness regarding this possibility in patients with positive genetic background.

Round 2

Reviewer 3 Report

The authors reported the case of MOGAD-transverse myelitis in a boy who was admitted at the hospital for bilateral motor deficit of the lower limbs and potentially associated with the covid19 vaccine. The manuscript is well written, with the review of the literature.

Author Response

Thank you very much for assessing our manuscript and for your positive comments. We hope to have fulfilled your previous requests, but please let us know if you consider that our manuscript needs further revisions.

Respectfully,

Lecturer Lorena Elena Melit, MD, PhD